# HIV-1 Circulating Recombinant Forms (CRFs) and Unique Recombinant Forms (URFs) in Israel, 2010–2018

**DOI:** 10.3390/v14091970

**Published:** 2022-09-06

**Authors:** Tali Wagner, Neta S. Zuckerman, Marina Wax, Rachel Shirazi, Yael Gozlan, Shirley Girshengorn, Rotem Marom, Ella Mendelson, Dan Turner, Orna Mor

**Affiliations:** 1Sackler Faculty of Medicine, Tel-Aviv University, Tel Aviv 6997801, Israel; 2Central Virology Laboratory, Ministry of Health, Sheba Medical Center, Ramat Gan 5262112, Israel; 3Tel-Aviv Sourasky Medical Center, Crusaid Kobler AIDS Center, Tel Aviv 6423906, Israel

**Keywords:** epidemiology, HIV-1 inter-subtype diversity, circulating recombinant forms (CRFs), transmitted drug resistance mutations (TDRM), HIV-1 drug resistance mutations (HIVdrm)

## Abstract

Monitoring HIV-1 circulating recombinant forms (CRFs) and unique recombinant forms (URFs) is important for disease surveillance. Recombination may affect prevention efforts and interfere with the diagnosis and treatment of HIV-1 infection. Here, we characterized the epidemiology of HIV-1 CRFs and URFs in Israel. Partial *pol* sequences from treatment naïve patients diagnosed in 2010–2018 were assessed using the recombinant identification program (RIP), the recombinant detection program (RDP5), and using the maximum-likelihood phylogenetic method, using 410 reference sequences obtained from the Los Alamos database. CRFs and URFs were identified in 11% (213/1940) of all sequenced cases. The median age at diagnosis was 38 (30–47) years, 61% originated from Israel, and 82% were male. The most common were CRF02_AG (30.5%), CRF01_AE (16.9%), and the more complex forms CRF01_AE/CRF02_AG/A3 (10.8%) and B/F1 (7%). A significant increase in their overall proportion was observed in recent years (8.1% in 2010–2012, 20.3% in 2016–2018, *p* < 0.001). This increase was most prominent in individuals carrying CRF02_AG (2.5% in 2010–2015, 9.8% in 2016–2018, *p* < 0.001). Men who have sex with men (MSM) was the most common risk group; however, those infected with the secondary recombinant CRF02_AG/A6 were mainly injecting drug users (IDUs). The most common resistance mutations were K103N (5/213, 2.3%) and E138A (18/213, 8.5%) in the reverse transcriptase. Only E138A was more frequent in the recombinants compared with the classic subtypes and was significantly associated with a specific secondary CRF, CRF02_AG/A4. We concluded that CRFs and URFs were mainly detected in Israeli-born MSM and that an increase in the overall proportion of such HIV-1 sequences could be observed in more recent years.

## 1. Introduction

HIV-1 molecular evolution continuously challenges treatment and prevention efforts [1]. Moreover, the increase in the proportions of the different HIV-1 inter-subtypes circulating in each country and within the different genome segments (*gag*, *pol*, and *env*) may challenge the development of an HIV subtype-specific vaccine [2]. HIV-1 recombination and molecular diversity may also impact transmission efficiency, pathogenesis, and may affect the performance of diagnostic tests and viral load measurements, as well as ART effectiveness and the impact of drug resistance mutations [3].

HIV-1 in Israel started with a subtype-B epidemic among MSM, followed by the introduction of subtypes C from Sub- Saharan Africa (SSA) (mainly by heterosexual transmission) in the 1980s and A from Eastern Europe (mainly by injecting drug users (IDUs)) in the 1990s. With the continuous waves of immigration from SSA, the most common subtype in those years was subtype C [4].

An analysis of the of HIV infection in 2010–2018 in Israel revealed that the epidemic is changing. Overall, 43.9% of the infections were subtype B, 25.2% subtype A (22.8% A6 2.4% with A1), and 17.1% subtype C [5]. However, all other infections with non-classic A, B, and C subtypes (13.8%) were not characterized thoroughly.

Various studies that reported global systematic analysis of HIV-1 molecular epidemiology demonstrated that proportions of HIV-1 recombinants (CRFs and URFs) changed over time, countries, and regions. The distribution of HIV-1 variants may be influenced by different factors, such as social transmission networks, transportation networks, and migration [1,6,7,8]. Thus, the prevalence of CRF02_AG was found to be the highest in Africa, subtype CRF01_AE is common in Asia and the Pacific, CRF07_BC in China, B/F1 in Latin America, and CRF63_02A6 in Eastern Europe and Central Asia [6,7,8,9]. Israel is a melting pot of immigrants from different countries; therefore, the continuous introduction of new HIV-1 subtypes and of recombinants could be expected.

Here, using partial *pol* sequences, we aimed to analyze the diversity of these HIV-1 CRFs and URFs and to compare the association of demographic and virological parameters between people living with HIV-1(PLHIV) harboring such recombinant viruses and those infected with HIV-1 B, C, A1, and A6 subtypes.

## 2. Materials and Methods

### 2.1. Identification of CRFs and URFs Using RIP, RDP5, and Phylogenetic Analysis

The Israeli HIV-1 database, which is located in the National HIV Reference Laboratory (NHRL) of the Ministry of Health (MoH) and in the HIV laboratory of the Sourasky Medical Center and contains HIV-1 sequences from treatment naïve PLHIV, >18 years of age, diagnosed in 2010–2018 (*n* = 1957), was searched for those with non-A, B, and C subtypes (Figure 1).

First, partial protease (PR, codons 4–99) and reverse transcriptase (RT, codons 38–247) sequences from samples suspected to contain HIV-1 recombinant forms (single sequence per patient, *n* = 270) were assembled using R script. Sequences shorter the 918 nucleotides (*n* = 17) were excluded. The resulting assembled 253 HIV-1 sequences were aligned (using Ali View version 1.27) [10] with 410 additional representative reference sequences covering all known CRFs and non CRFs of group M, including sequences obtained by Blast analysis related to the samples analyzed herein. These were obtained from the Compendium of the Los Alamos database (www.hiv.lanl.gov/content/sequence/HIV/CRFs/CRFs.html, (accessed on 10 August 2022), and HIV BLAST (https://www.hiv.lanl.gov/content/sequence/BASIC_BLAST/basic_blast.html), accessed on 10 August 2022. Recombinants were defined with RIP (Recombinant Identification Program, available at www.hiv.lanl.gov/content/sequence/RIP/RIP.html, accessed on 10 August 2022) and with RDP (Recombination Detection Program) version 5.5 [11]. CRFs were determined according to the guidelines available at (www.hiv.lanl.gov/content/sequence/HIV/CRFs/crfs.comp, accessed on 10 August 2022). All of these sequences were also explored with Mega X software (http://www.megasoftware.net, accessed on 10 August 2022) using the maximum-likelihood phylogenetic method and the General time-reversible model, with the proportion of invariable sites and gamma plus invariant sites-distributed rate heterogeneity (GTR + G + I model) based on the results of JModelTest [12] version 2.1.9. A maximum likelihood phylogenetic tree was constructed with branch support assessed from 1000 bootstrap replicates and visualized with Fig Tree version 1.4.4 (http://tree.bio.ed.ac.uk/software/figtree/, accessed on 10 August 2022). A phylogenetic tree was used not only for subtype identification, but also to assist in assessing borderline RIP and RDP results. Following the analysis of all of the sequences with these three tools, sequences with no evidence of recombination (*n* = 40) were removed from further analysis, yielding a final set of 213 HIV-1 recombinants. The results of these analyses and the detailed list of all 213 sequences of HIV-1 recombinants is provided in Appendix A. Recombination breakpoints (according to HXB2 genome positions) were obtained using RDP5 and Simplot v3.5.1 software. Mosaic patterns were generated using the Recombinant HIV-1 Drawing Tool (https://www.hiv.lanl.gov/content/sequence/DRAW_CRF/recom_mapper.html, accessed on 10 August 2022). An analysis of all drug resistance mutations was performed using the Stanford HIVdb algorithm version 9.0 [13]. Transmitted drug resistance mutations (TDRM) were defined according to Bennett et al. [14]. We used both these algorithms (HIVdb and TDRM analysis) to enable a comparison of our results to published reports, as it was already indicated that there is currently no consolidation in the reported mutations that are in use to determine the prevalence data [15].

### 2.2. Comparison between Patients Carrying Recombinant viruses and Patients Infected with Non CRFs

To compare the characteristics of the individuals infected with subtypes (A, B, and C) with those with recombinant subtypes, a random sample (50%) of individuals selected from all treatment naïve HIV-1 positives diagnosed in 2010–2018 was utilized. Patients with HIV-1 viral load <3 log c/mL at diagnosis were excluded, resulting in 3432 eligible individuals. Random selection was performed following the stratification of these 3432 cases to three groups of years of diagnosis (2010–2012, 2013–2015, and 2016–2018). Only patients with sequences (928/1716) were included in the final comparison. Overall, 928 individuals with HIV-1 sequence data were selected, of which 808 infected with A, B, and C subtypes and 120 with recombinant viruses (Appendix A). Subsequently, a comparison of patient’s characteristics, demographic (sex, age, region of birth, available travel history, and possible mode of HIV-1 transmission), and clinical (HIV-1viral load in Logs copies/mL, CD4 cell counts in cells/mm^3^) data were retrieved. Country of birth was categorized based on https://www.unaids.org/en/regionscountries/regions (accessed on 10 August 2022) definitions and as previously described (5): SSA (Ethiopia, Ghana, Sudan, Nigeria, Congo, and Ivory Coast); Eastern Europe and Central Asia, EEU/CA, (Russian Federation, Ukraine, Azerbaijan, and Uzbekistan); Western and Central Europe and North America, WCEU/NA (United Kingdom, Romania, Holland, and the United States); Latin America, LA (Argentina, Brazil, Peru, Uruguay, and Venezuela); Asia and the Pacific, AP (Philippine, Thailand, and India); the Middle East and North Africa, ME/NA (Jordan, Tunisia, Yemen, Morocco, and Algeria); and Israel.

### 2.3. Statistical Analysis and Ethical Approval

Descriptive statistics was used to assess the study cohort. Variables with non-Gaussian abnormal distribution were expressed by median and interquartile range, and the Mann–Whitney test was performed to test the quality of the means of several distributions. Categorical variables were expressed by frequencies and compared using the chi-squared test. Statistical analysis was performed using IBM SPSS statistics version 20.

The study was approved by the institutional review boards of the Sheba and Sourasky Medical Centers. All data were anonymized, cross-matched, and coded as dual-encoding on demand. A consent waiver was obtained for the study.

## 3. Results

### 3.1. Characteristics of HIV-1 Recombinants

An analysis of all of the sequenced samples from individuals diagnosed in 2010–2018 identified 213 PLHIV infected with recombinant viruses. Figure 2 presents the phylogenetic analysis of all these recombinants. Nine main clusters (>3 individuals, bootstrap >0.8%) were determined. The characterization of the sequences within the main nine clusters and the pattern of representative sequences from these clusters are shown in Appendix A.

Figure 3 shows the proportion of different HIV-1 recombinants identified. The most common were CRF02_AG (30.5%) and CRF01_AE (16.9%). Other recombinants identified were B/F1 (7%), A6/B (3.8%), C/A1 (2.8%), B/D (2.3%), and B/C (1.9%). Secondary recombinants were identified in 26% of all recombinant forms and included CRF01_AE/CRF02_AG/A3 (10.8%), CRF02_AG/A4 (6.6%), CRF02_AG/B (2.8%), CRF02_AG/G (2.8), CRF01_AE/B (1.4%), and CRF02_AG/A6 (1.4%).

Table 1 demonstrates the characteristics of PLHIV infected with recombinant viruses. The median age at diagnosis was 38 (30–47) years, 61% were born in Israel, and 82% were male. The median CD4 and HIV-1 RNA levels were 389 (214–511) cells/mm^3^ and 4.9 (4.3–5.4) log copies/mL, respectively. MSM was the most common risk group (58.5%) identified, followed by heterosexual contacts (35.6%). No significant changes in the characteristics of the patients with recombinant viruses (e.g., in the proportion of risk groups, median viral loads, and median CD4 counts) was observed in the study period. In the largest CRF groups (CRF02_AG and CRF01_AE), MSM and those infected through heterosexual contacts comprised a similar fraction (*p* = 0.082). Individuals infected with the secondary recombinant CRF02_AG/A6 were either injecting drug users or heterosexuals (66% and 33%, respectively).

The majority of PLHIV harboring CRFs and URFs (61%) were born in Israel. A travel history to HIV-1 endemic countries was reported for few cases (7/127 Israelis). Indeed, for these individuals as well as for those born in such endemic areas, an association between the HIV-1 recombinant and the country of infection could be identified. Thus, PLHIV born in or travelled to Africa were most often infected with CRF02_AG (16/25, 64%, *p* < 0.001). The CRF01_AE subtype characterized HIV-1 positives who were infected in Asia and the Pacific (9/12, 75%, *p* < 0.001). Most of those with B/F1 virus were infected in Latin America (4/6, 66.7%, *p* < 0.001). A phylogenetic analysis demonstrated that most (70%) of PLHIV infected outside Israel were not clustered. In addition, there were no significant differences in the characteristics of PLHIV that were clustered and PLHIV who were not in any cluster. Large clusters (cluster IV- CRF02_AG/CRF01_AE/A3 and cluster V- CRF02_AG/A4) were characterized mainly by Israeli born MSM who were infected in more recent years (Figure 2 and Appendix A). 

The resistance mutation analysis revealed that mutations in the protease were very rare, accounting for less than 1.5% of cases. K103NS and E138A were the most abundant NNRTI mutations (6/213, 2.8% and 18/213, 8.5%, respectively). E138A was the only mutation significantly associated (*p* < 0.001) with a specific secondary CRF, CRF02_AG/A4 (Appendix A).

### 3.2. Comparison between PLHIV Carrying Recombinant Viruses to those Infected with Classic Subtypes

A random sample of all individuals diagnosed in 2010–2018 was selected (Appendix A) to enable a comparison of the characteristics of those infected with A, B, or C subtypes to those with HIV-1 recombinant viruses described herein. Table 2 compares of the characteristics of these 928 randomly selected individuals.

Significantly more males were identified with recombinant viruses (*p* = 0.038). While the proportion of PLHIV who originated from EEU/CA was larger among individuals infected with classic subtypes, among PLHIV infected with recombinant viruses, the proportion of Israeli born individuals was higher (*p* = 0.023). MSM was the most prominent transmission group in recombinants (*p* = 0.011). While the median CD4 counts were higher in individuals who were infected with recombinant viruses compared with classic subtypes (*p* = 0.043), the median HIV-1 viral loads were similar in these two groups (*p* = 0.303). Resistance analysis revealed that the magnitude of resistance mutations was similar between those carrying recombinant viruses and others who were infected with classic subtypes (8.3% vs. 14.2%, respectively, *p* = 0.085 for TDRM and 19.2% vs. 24.3%, respectively = 0.249 for HIVdrm). The only significant differences in the proportions of resistance mutations was for E138A, which was more frequent in the recombinants compared with the non CRFs (7.5% vs. 2.9%, respectively, *p* = 0.028).

When years of diagnosis were compared, a significant increase in the overall proportion of CRFs was observed in recent years (from 8.1% in 2010–2012 to 20.3% in 2016–2018, *p* < 0.001, Figure 4). This significant increase in the proportion of CRFs was particularily prominent in 2016–2018, among those who were harboring CRF02_AG (from 2.5% in 2010–2015 to 9.8% in 2016–2018, *p* < 0.001).

## 4. Discussion

With the continuous global spread of HIV-1, the constant viral evolution and increase in the number and proportion of recombinant forms is an ongoing concern. In Israel, 213 of all *pol* sequences derived from PLHIV identified in 2010–2018 were found to belong to recombinant viruses. As in other regions in Europe [16], the overall proportion of recombinants increased over the study period. Two main types of recombinants were identified: CRF02_AG, which was the most common form (30.5%), increasing particularly in recent years, and CRF01_AE, which was identified 16.9% of all recombinants. Indeed, globally, CRF02_AG is the most common recombinant form comprising 33.9% of all identified recombinant infections [1]. In the same report, CRF01_AE exhibited 23% of all globally circulating recombinant forms [1]. In addition to these two recombinant forms, secondary recombinants or more complex forms of the recombinants, such as those that combine CRF02_AG and CRF01_AE, have been found, suggesting a high genetic variability and extensive heterogeneity of the virus [17]. Indeed, in Africa, where the epidemic is ongoing mainly by heterosexual transmission, the evolutionary history of CRF02_AG combined with the high prevalence and high replication capacity of this form allows this recombinant to produce more complex forms [6].

Most of those newly infected with the recombinant virus (66.1%) were PLHIV born in Israel. For a minority of these individuals, information on infection in endemic regions was available and could partially explain the introduction of such HIV-1 recombinants in Israel. Indeed, data on country of infection are important in order to understand the epidemiology and spread of HIV recombinants for regional and global surveillance. Active circulation of HIV-1 recombinants in Israel could not be determined, as the characteristics of clustered individuals were not significantly different to those who were not in any cluster.

While 61.5% of those infected with recombinants were MSM, this HIV-1 transmission group comprised only 48.5% of those infected with classic subtypes (*p* = 0.011). The high genetic diversity of HIV-1 already observed in MSMs [18] and the possibility of cross-ethnic transmission [4,19], potentially, could increase the spread of recombinants with more complex forms of the virus in the future. While the sexual mode of transmission was the most common among all CRF groups, CRF02_AG/A6 was mainly identified in IDUs. The identification of CRF02_AG/A6 among this risk group could be explained by the transmission of this recombinant in Uzbekistan, Kazakhstan, and Russia, and the further introduction of this virus in Israel by immigrants from EEU/CA [1,8].

Although the median CD4 cell counts upon diagnosis were slightly higher in PLHIV with recombinant viruses compared with those with non CRFs (median of 418 versus 323 cells/mm^3^, respectively), it was still considered abnormal [20]. Efforts to improve universal HIV free of charge testing and raising the awareness about this disease among risk groups and healthcare providers are still needed.

It has already been shown that the polymorphic accessory mutation E138A, which affects etravirine and rilpivirine (RPV), was the most prominent in non-classic subtypes [5]. Here, we show that this mutation was significantly associated with a specific secondary CRF, CRF02_AG/A4 recombinant. In such cases, where the E138A mutation is present, the use of a cabotegravir (CAB) CAB/RPV should be considered as a relative contraindication [13]. Such data emphasize the continuous need for baseline resistance, especially in the current era of treatment simplification, and of shifting to dual therapy regimens that contain RPV.

Our study had several inherent limitations. First, only partial *pol* genome sequences were available for the analysis; therefore, there may be an underestimation of the recombination events. In addition, our sequences did not include an integrase region, as treatment based on integrase inhibitors in first-line therapy was mainly introduced in 2016. These limitations have been observed in other studies as well. For example, Hemelaar et al. [1] used partial *pol* sequences to describe the global proportion of CRFs in recent years. This inherent limitation may also be considered as an advantage when assessing the diversity of HIV-1, because there are many more *pol* sequences in the databases compared with complete genome sequences [21]. However, in the future, we also plan to obtain a full genome sequence for some of these recombinants, especially for those that could not be assigned clearly to a known CRF. A reporting bias may be also present, such as reporting sexual behavior, drug injection, or previous treatment exposure. To partially overcome these obstacles, data were compared and cross-matched before the analysis, and only observations with HIV-1—RNA > 3 log copies/mL (considered to be treatment naïve) were selected for comparison analysis between recombinants and between non CRFs. Despite these limitations, the application of two recombination detection methods, as well as the robust phylogenetic analysis enabled the identification of a variety of HIV-1 recombinants.

In conclusion, our study documents an increase in the proportion of recombinant viruses among HIV-1 infected individuals, particularly in recent years (2016–2018) in Israel. Furthermore, the local circulation of HIV-1 recombinants is mainly by the MSM. These findings highlight the need for strengthening the monitoring efforts at regional and global levels and implementing effective means to reduce the spread of recombinant forms.

## Figures and Tables

**Figure 1 viruses-14-01970-f001:**
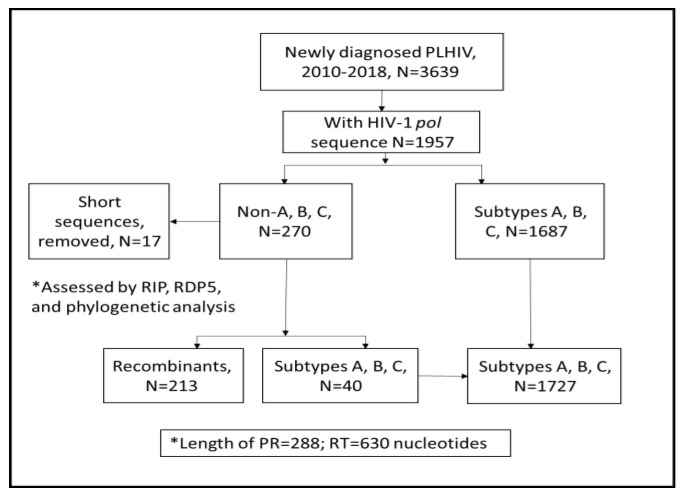
Identification of HIV-1 CRFs and URFs (recombinants) in 2010–2018. All PLHIV newly diagnosed in 2010–2018 (N = 3639) for whom sequencing data were available (*n* = 1957) were included in the analysis as described in the Materials and Methods section. The RIP, RDP5, and phylogenetic analysis revealed that 213 of these sequences are non A, B, and C. RIP—Recombinant Identification Program; RDP5—Recombinant Detection Program (version 5.5).

**Figure 2 viruses-14-01970-f002:**
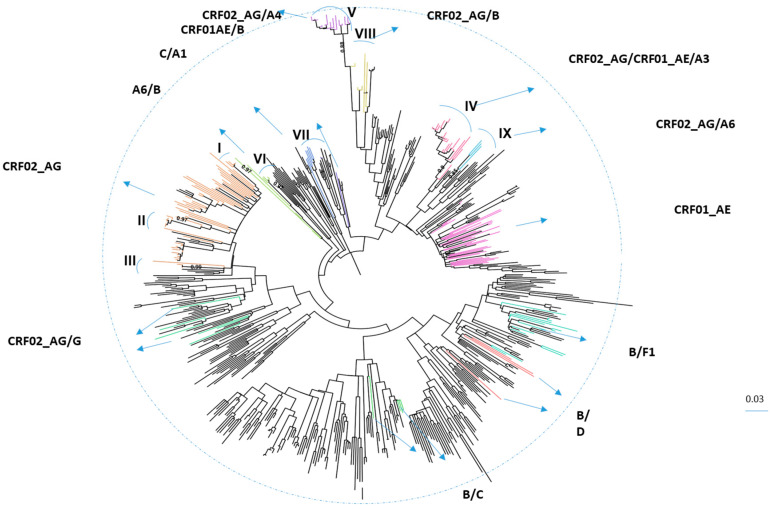
Phylogenetic analysis of HIV-1 recombinants in Israel, 2010–2018. Phylogenetic tree of the *pol* region of HIV-1 from 213 patients was generated by the Mega X and constructed using the maximum likelihood (ML) with GTR + G + I model. Included were also 410 reference sequences (pure and recombinant forms) obtained from the Los Alamos database. The tree was visualized in Fig Tree version 1.4.4. HIV-1 recombinants are colored as follows: CRF02_AG in pink, CRF01_AE in purple, CRF01_AE/B in violet, CRF02_AG/CRF01_AE/A3 in rose, B/F1 in turquoise, CRF02_AG/A4 in lilac purple, A6/B in green, C/A1 in blue, CRF02_AG/B in gold, CRF02_AG/A6 in magic mint, CRF02_AG/G in Caribbean green, B/C in green, and B/D in terrain. Reference sequences are in black. The main clusters were determined according to the bootstrap values (>0.8) and number of individuals (>2) in each cluster and named with the Roman letters (I–IX) and indicated with the blue splines. The bootstraps of each cluster were: I. 0.97; II. 0.97; III. 0.99; IV. 0.9; V. 0.88; VI. 0.97; VII. 1; VIII. 1; IX. 0.85.

**Figure 3 viruses-14-01970-f003:**
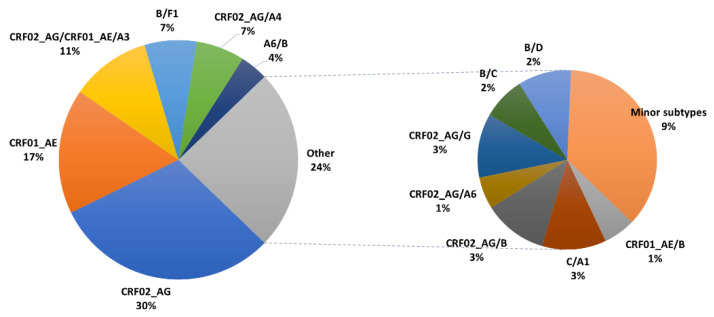
Proportion of HIV-1 CRFs and URFs in Israel, 2010–2018. Here, the large pie shows the distribution of the most prevalent ones, while the small one shows the distribution of less common forms of recombinants.

**Figure 4 viruses-14-01970-f004:**
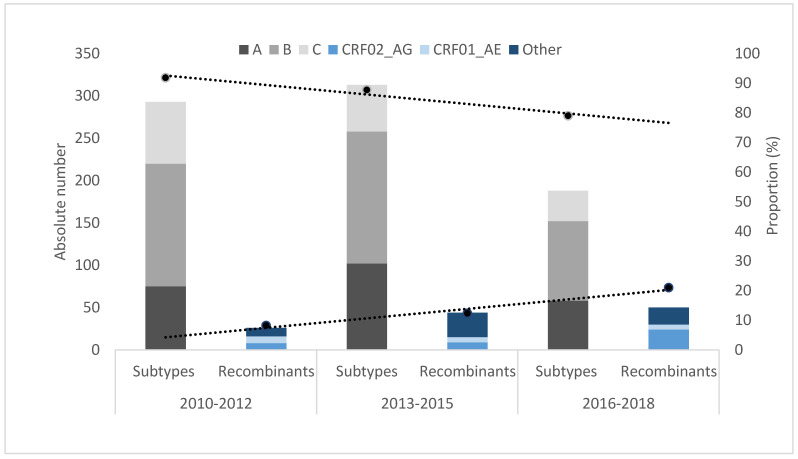
HIV-1 subtypes versus recombinant forms by years of diagnosis 2010–2012, 2013–2015, and 2016–2018. Number of HIV-1 subtypes is presented by black-grey columns (subtype A—dark grey, B—lighter grey, and C—very light grey). Recombinant forms are in blue colors (CRF02_AG in light blue, CRF01_AE in lighter blue, and all other recombinants in dark blue). Black dotted lines denote the trend in the proportion of HIV-1 subtypes and recombinants.

**Table 1 viruses-14-01970-t001:** Characteristics of patients harboring HIV-1 CRFs and URFs.

Recombinants	CRF02_AG	CRF01_AE	CRF02_AG/ CRF01_AE/A3	B/F1	CRF02_AG/A4	A6/B	Other	All
N (%)	65 (30.5)	36 (16.9)	23 (10.8)	15 (7)	14 (6.6)	8 (3.8)	52 (24.4)	213 (100)
Age, Year, Median (IQR)	37 (30–42)	35 (26–47)	47 (38–54)	40 (32–45)	28 (24–43)	28 (23–51)	41 (35–50)	38 (30–47)
Male, N (%)	45 (69.2)	31 (86.1)	22 (95.7)	12 (80)	14 (100)	8 (100)	43 (82.7)	175 (82.5)
Place of Birth								
SSA	14/63 (22.2)		2/23 (8.7)		2/14 (14.3)		5/50 (10)	23 (11)
EEU/CA	10/63 (15.9)	3/36 (8.3)	2/23 (8.7)	1/15 (6.7)	5/14 (35.7)	1/8 (12.5)	9/50 (18)	31 (14.8)
WCEU/NA	6/63 (9.5)	1/36 (2.8)	1/23 (4.3)				1/50 (2)	9 (4.3)
Israel	28/63 (44.4)	27/36 (75)	15/23 (65.2)	10/15 (66.7)	6/14 (42.9)	7/8 (87.5)	34/50 (68)	127 (60.8)
Other	5/63 (7.9)	5/36 (13.9)	3/23 (13)	4/15 (26.7)	1/14 (7.1)		1/50 (2)	19 (9.1)
Transmission Groups								
MSM	29/63 (46)	19/34 (55.9)	18/23 (78.3)	10/13 (76.9)	11/14 (78.6)	7/8 (87.5)	26/50 (52)	120 (58.5)
HETERO	28/63 (44.4)	14/34 (41.2)	5/23 (21.7)	2/13 (15.4)	3/14 (21.4)	1/8 (12.5)	20/50 (40)	73 (35.6)
IDU	6/63 (9.5)	1/34 (2.9)		1/13 (7.7)			4/50 (8)	12 (5.9)
CD4, *n*/N(%)	55/65 (28.4)	35/36 (18)	23/23 (11.9)	15/15 (7.7)	13/14 (6.7)	8/8 (4.1)	45/52 (23.2)	194 (91.1)
CD4(cells/mm^3^), Median (IQR)	383 (216–515)	373 (204–609)	418 (216–514)	408 (243–473)	497 (415–739)	343 (287–749)	350 (161–451)	389 (214–511)
HIV-1 VL *n*/N (%)	61/65 (29.8)	36/36 (17.6)	23/23 (11.2)	15/15 (7.3)	13/14 (6.3)	8/8 (3.9)	49/52 (23.9)	205 (96.2)
HIV-1 VL, Median (IQR)	4.9 (2.2–5.4)	4.9 (4.3–5.3)	4.9 (4.5–5.3)	5.5 (3.8–6.1)	4.5 (4–5.2)	4.5 (4–4.9)	4.9 (4.6–5.5)	4.9 (4.3–5.4)

HIV-1 VL—viral load (Log copies/mL); SSA—Sub- Saharan Africa; EEU/CA—Eastern Europe, and Central Asia; WCEU/NA—Western and Central Europe and North America; MSM—men who have sex with men; Hetero—heterosexual contacts; IDU—injecting drug users. Data are presented as n (%) or median (IQR); IQR—interquartile range.

**Table 2 viruses-14-01970-t002:** Comparison of PLHIV diagnosed in 2010–2018 infected with A, B, and C subtypes to those with CRFs and URFs (recombinant forms).

	Subtypes (N = 808)	Recombinant Forms (N = 120)	*p*-Value
Age at diagnosis (years)	N = 802 (87)	N = 120 (13)	0.936
median (IQR) (*n* = 922)	38 (30–45)	39 (29–45)	
Sex, (*n* = 923, 99.5%)	N = 803 (87)	N = 120 (13)	0.038
Female	196 (24.4)	19 (15.8)	
Male	607 (75.6)	101 (84.2)	
Place of birth, (n = 893, 96.2%)	N = 775 (86.8)	N = 118 (13.2)	
SSA	125 (16.1)	14 (11.9)	
EEU/CA	213 (27.5)	17 (14.4)	
WCE/NA	25 (3.2)	3 (2.5)	
Israel	389 (50.2)	78 (66.1)	0.023
LA	7 (0.9)	2 (1.7)	
AP	7 (0.9)	2 (1.7)	
ME/NA	9 (1.2)	2 (1.7)	
Transmission Groups, (*n* = 899, 96.9%)	N = 782 (87)	N = 117 (13)	
MSM	378 (48.3)	72 (61.5)	
Hetero	288 (36.8)	37 (31.6)	0.011
IDU	116 (14.8)	8 (6.8)	
CD4 (cells/mm^3^), (N = 849, 91.5%)	N = 741 (87.3)	N = 108 (12.7)	
median CD4 (IQR)	323 (175–519)	418 (209–509)	0.043
HIV-1—RNA (Log copies/mL),	N = 772 (87.2)	N = 113 (12.8)	
(N = 885, 95.4)			
median viral load (IQR)	4.8 (4.2–5.4)	4.9 (4.3–5.5)	0.303
TDRM Bennet (N = 906, 97.6%)TDRM, *n* (%)	N = 786 (86.8)	N = 120 (13.2)	
	112 (14.2)	10 (8.3)	0.085
HIVdrm ECDC (N = 906, 97.6%)HIVdrm, *n* (%)	N = 786 (86.8)	N = 120 (13.2)	
	191 (24.3)	23 (19.2)	0.249

SSA—Sub-Saharan Africa; EEU/CA—Eastern Europe and Central Asia, WCEU/NA-West/Central Europe/North America; LA—Latin America; AP-Asia and Pacific; ME/NA—Middle East and North Africa MSM- Men who have sex with men; Hetero- Heterosexual contacts; IDU—Injecting drug users. TDRM- Transmitted drug resistance mutations. HIVdrm—HIV-1 drug-resistance mutations. ECDC—European Centre for Disease Prevention and Control. Data are presented as n (%) or median and IQR—interquartile range. Significance of differences was measured using the Chi-squared test (or Fisher’s Exact test) for categorical variables like sex, place of birth, transmission groups, TDRM Bennet, HIVdrm ECDC or Mann–Whitney test for continuous variables such as age at diagnoses, CD4, and HIV-1–RNA.

## Data Availability

Raw data can be provided upon request from the authors.

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
