# Peer review of "HIV-1 Circulating Recombinant Forms (CRFs) and Unique Recombinant Forms (URFs) in Israel, 2010–2018"

_viruses, 2022, doi:10.3390/v14091970_

Round 1

Reviewer 1 Report

In this manuscript, Wagner et al. report the characterization of the diversity and epidemiology of HIV-1 inter-subtype recombinants in Israel from 2010 to 2018 among newly diagnosed PLHIV. They report 11% (213/1940)  of sequences were recombinant forms (CRFs) with an increase in frequency of CRFs over the studied period. Multiple types of CRFs and various complex CFRs were identified amongst the studied individuals. The authors identify MSM as a group with higher risk of presenting CRFs than other risk groups. The study provides information regarding the evolution of HIV-1 in Israel, noting the increased frequency of recombinant forms (CRFs) over the studied period in the newly diagnosed population.

Overall, this is an interesting manuscript, and the research and interpretation are adequate.

The Reviewer found several issues in the manuscript and would like to see them addressed by the Authors before this manuscript is ready for publication.

 Specific comments and suggestions:

A.    INTRODUCTION

1. In general, the Introduction could use additional content to put into context the significance of this study for Israel and its effort to contain the HIV epidemic in their region. The Authors should add to the references with recent literature (5 years or less).

2. Authors should elaborate whether there are additional studies done describing the circulating HIV-1 virus in Israel and regarding the presence of CRFs in Israel in addition to their own citation, or, if this manuscript is describing the first characterization of CRFs in Israel. The Reviewer would like to see these points clearly stated.

3. On line 62, authors refer to “pure HIV- 62 1 B, C, A1 and A6 subtypes” however this is confusing as the authors did not study complete genomes to determine there is no recombination in other parts of the genome. As in the current study, the Authors analyzed partial pol sequences and therefore can indicate the virus is single subtype in their pol sequences but should not identify them as “pure” subtypes.

4. Line 47-revise for grammar and/ or typos. Remove extra “to.” 

B.    MATERIALS AND METHODS

1.  On line 69, Authors mention sequences “suspected to be recombinants”. Please explain, how did the authors identify these sequences as “suspect” prior to the sequence analysis which is the object of this manuscript? Or did the Authors have a priori knowledge of which sequences were recombinants and they are further characterizing them? Please eliminate the term “suspected/ suspect” if there was no prior knowledge of their state as recombinants. Otherwise make this point clear in the manuscript.

2.  Line 104, once again, Authors refer to “pure” subtypes, however without complete genome analysis this would be inaccurate. Authors could simply indicate “subtypes (A, B, C) “ vs. “recombinant subtypes” and that clearly conveys the meaning of their analysis without implying a deeper analysis (whole genome) of their sequences to be able to determine they are actually “pure” subtype. Otherwise, justify why the use of the terminology of “pure subtypes” without whole genome analyses would be accurate.

3.  The section “Comparison between patients…”, paragraph from lines 103 to 117, is not clear. The Reviewer found confusing the selection process as described. Clarify, how if 3,432 individuals were eligible and the sample taken was 50% of those, just how the final analysis ends up with 928 individuals being selected for analysis which is the 25% of those eligible? Or are the Authors referring to the 1957 individuals with sequences available? Please, verify this paragraph and clarify this point.

4.  Authors, give explanation why not work the analysis including the 1,727 “pure” subtype and the 213 recombinant sequences that were identified (mentioned in Fig. 1). Seems that the Authors would be losing valuable information from the variety of recombinants by doing a 50% sampling. Please explain why was that sampling was done in the first place when the Authors are using descriptive statistics and pairwise comparisons? Please elaborate on these points.

The Reviewer found minor issues with Tables and Figures that affect understanding the data presented and would like to see them addressed.

A.    MATERIALS AND METHODS

 1.     Figure 1.  Consider simplifying the caption, most of the information is found elsewhere in the manuscript. Figure would be easier to read using by using black lettering and lines.

B.    RESULTS

1.     Figure 2.  The labels inside the pie chart are difficult to read. The Authors should avoid the use of dark lettering on dark colored backgrounds and font size is too small for effortless reading. Please make the corresponding adjustments. Please, keep placement of labels consistent, either inside chart or outside.

2.     Table 1.  Table is exceedingly difficult to read and understand due to extremely small lettering. The Table contains a large amount of data which makes it look cluttered. It would benefit of being make simpler or divided in parts. Authors should consider presenting part of these data as figures/charts and include the full table as a supplementary file.

3.     Table 1 and 2.

a. The Reviewer did not understand why the Authors lump North America together with Western and Central Europe. The Reviewer suggests having North America categorized separate as done for “Latin America” (see next point) which geographically makes sense.

b. The Authors use the category “Latin America” however, that is more a cultural and linguistical regional division and not a geographical one. The Reviewer suggests following the geographical division for clarity. Instead of using “Latin America,” please describe this category as “Central/South America” or “Central America” and “South America”  whichever is the case. Are the specific countries known? Are the Caribbean countries included?

4. Table 2

a.    It is exceedingly difficult to read as formatted. Since Column 1 is unlabeled and line items below are centered, it is difficult to identify which lines are divisions and which ones are sub-divisions of these. The Reviewer suggests changing the formatting for this Table. As an example, place the “divisions” (i.e., age, sex, place of birth) towards the left margin and right-indent the subdivisions (i.e., female, male) but also aligned towards the left. The same should be done for the other columns. This would direct the line of vision making it easier to understand specially when each division has different N values.

b.  Also, it is very confusing the label for columns 2 and 3 indicating “N=808” and “N=120”, respectively. It takes too much time to decipher those numbers have nothing to do with the lines below since all have different “N” numbers. Shouldn’t these “Ns” be written as “n”? aren’t they a sample of the totality of sequences evaluated? Please review and make the corresponding changes to the Table.

c.     Please check that the location of the “p values” are in the correct line. For example,  in the line “Sex…”, the p value is placed on same line of “female” yet in the text (line 183) it is described as “more males were identified with recombinant viruses (p=0.038).” Observing Table 2 it seems to the Reviewer that the p value refers to “females” and not “males”. Same issue for “ Place of birth…” where the p value is placed in that same line yet in the text (line 186) refers to “Israeli born individuals”  yet in the table “Israel” is placed 4 lines down from where the p value is. Please review for clarity and make corresponding changes to  the Table.

2.  Figure 3. Consider simplifying the caption as the method and results are found elsewhere in the text.

Author Response

 Response to Reviewer 1 Comments

  1. INTRODUCTION
  1. In general, the Introduction could use additional content to put into context the significance of this study for Israel and its effort to contain the HIV epidemic in their region. The Authors should add to the references with recent literature (5 years or less).

Our reply: We revised the introduction and added more on the significance of this study. As no new and relevant Israeli molecular epidemiologic studies were published by others in the last 5 years, we added some background info on earlier local data.

The revised section is as follows: "HIV-1 in Israel started with a subtype-B epidemic among MSM and following by introductions of subtypes C from Sub Saharan Africa (SSA) (mainly by heterosexual transmission) in the 1980s and A from Eastern Europe (mainly by injecting drug users, IDU's) in the 1990s. With the continuous waves of immigration from SSA, the most common subtype in those years was subtype C. (4).

Analysis of the of HIV infection in 2010-2018 in Israel revealed that the epidemic is changing. Overall, 43.9% of the infections were with subtype B, 25.2% with subtype A (22.8% A6 2.4% with A1) and 17.1% with subtype C (5). However, all other infections with none-classic A, B, C subtypes (13.8 %) were not characterized thoroughly".

  1. Authors should elaborate whether there are additional studies done describing the circulating HIV-1 virus in Israel and regarding the presence of CRFs in Israel in addition to their own citation, or, if this manuscript is describing the first characterization of CRFs in Israel. The Reviewer would like to see these points clearly stated.

Our reply: Thank you. This study summarizes the first characterization of CRFs in Israel. Please see our previous reply for this revised section.

  1. On line 62, authors refer to “pure HIV- 62 1 B, C, A1 and A6 subtypes” however this is confusing as the authors did not study complete genomes to determine there is no recombination in other parts of the genome. As in the current study, the Authors analyzed partial polsequences and therefore can indicate the virus is single subtype in their pol sequences but should not identify them as “pure” subtypes.

Our reply: Thank you. This was corrected. The term Pure subtypes was removed. Also we revised as follows: "Here, using partial pol sequences we aimed to analyze the diversity of these HIV-1 recombinants and compare the association of demographic and virological parameters between people living with HIV-1(PLHIV) harboring recombinant viruses and those infected with HIV-1 B, C, A1 and A6 subtypes." 

  1. Line 47-revise for grammar and/ or typos. Remove extra “to.” 

Our reply: Thank you, done.

  1. MATERIALS AND METHODS
  2. On line 69, Authors mention sequences “suspected to be recombinants”. Please explain, how did the authors identify these sequences as “suspect” prior to the sequence analysis which is the object of this manuscript? Or did the Authors have a priori knowledge of which sequences were recombinants and they are further characterizing them? Please eliminate the term “suspected/ suspect” if there was no prior knowledge of their state as recombinants. Otherwise make this point clear in the manuscript.

Our reply: Thank you. Previous study demonstrated that the Israeli database contains about ~14% of sequences related to non subtypes A, B, C. In this report we focused on these sequences. Following your remark we have removed the term "suspected".

  1. Line 104, once again, Authors refer to “pure” subtypes, however without complete genome analysis this would be inaccurate. Authors could simply indicate “subtypes (A, B, C) “vs. “recombinant subtypes” and that clearly conveys the meaning of their analysis without implying a deeper analysis (whole genome) of their sequences to be able to determine they are actually “pure” subtype. Otherwise, justify why the use of the terminology of “pure subtypes” without whole genome analyses would be accurate.

Our reply: We agree. Therefore, following your remarks, we have revised this important issue all through this paper and removed the term "pure" and used subtypes instead throughout the paper.

  1. The section “Comparison between patients…”, paragraph from lines 103 to 117, is not clear. The Reviewer found confusing the selection process as described. Clarify, how if 3,432 individuals were eligible and the sample taken was 50% of those, just how the final analysis ends up with 928 individuals being selected for analysis which is the 25% of those eligible? Or are the Authors referring to the 1957 individuals with sequences available? Please, verify this paragraph and clarify this point.

Our reply: Thank you. Israeli database contains about~ 54% of sequenced cases in 2010-2018. (Other ~45% were not sequenced). To avoid selection bias of only those who were sequenced we randomly sampled 50% of all cased stratified by years of diagnosis (2010-2012,2013-2015,2016-2016).  We clarified this issue in the text and prepared a flow diagram aiming to better describe our approach.

The text was revised as follows:

"To compare the characteristics of individuals infected with subtypes (A, B, C) to those with recombinant subtypes, a random sample (50%) of individuals selected from all treatment naïve HIV-1 positives diagnosed in 2010-2018 was utilized. Patients with HIV-1 viral load <3 log c/ml at diagnosis were excluded, resulting in 3432 eligible individuals. Random selection was performed following stratification of these 3432 cases to three groups of years of diagnosis (2010-2012, 2013-2015, 2016-2018). Only patients with sequences (928/1716) were included in the final comparison. Overall, 928 individuals with HIV-1 sequence data were selected, 808 infected with A, B, C subtypes and 120 with recombinant viruses (Figure S2)."

  1. Authors, give explanation why not work the analysis including the 1,727 “pure” subtype and the 213 recombinant sequences that were identified (mentioned in Fig. 1). Seems that the Authors would be losing valuable information from the variety of recombinants by doing a 50% sampling. Please explain why was that sampling was done in the first place when the Authors are using descriptive statistics and pairwise comparisons? Please elaborate on these points.

Our reply: Please see our reply to previous point

  1. MATERIALS AND METHODS
  2. Figure 1.  Consider simplifying the caption, most of the information is found elsewhere in the manuscript. Figure would be easier to read using by using black lettering and lines.

Our reply: Thanks, corrected.

  1. RESULTS
  2. Figure 2.  The labels inside the pie chart are difficult to read. The Authors should avoid the use of dark lettering on dark colored backgrounds and font size is too small for effortless reading. Please make the corresponding adjustments. Please, keep placement of labels consistent, either inside chart or outside.

Our reply: Thanks, corrected.

  1. Table 1.  Table is exceedingly difficult to read and understand due to extremely small lettering. The Table contains a large amount of data which makes it look cluttered. It would benefit of being make simpler or divided in parts. Authors should consider presenting part of these data as figures/charts and include the full table as a supplementary file.

Our reply: Thanks, this Table was simplified in the revised version.

  1. Table 1 and 2.
  2. The Reviewer did not understand why the Authors lump North America together with Western and Central Europe. The Reviewer suggests having North America categorized separate as done for “Latin America” (see next point) which geographically makes sense.

Our reply: Thank you.

Birthplace was categorized based on https://www.unaids.org/en/regionscountries/regions definitions. There, Western and Central Europe are categorized with North America, WCEU/NA (United Kingdom, Romania, Holland, and United States)

  1. The Authors use the category “Latin America” however, that is more a cultural and linguisticalregional division and not a geographical one. The Reviewer suggests following the geographical division for clarity. Instead of using “Latin America,” please describe this category as “Central/South America” or “Central America” and “South America” whichever is the case. Are the specific countries known? Are the Caribbean countries included?

Our reply: Please, see previous reply. The text in the revised version reads as follows:

Country of Birth was categorized as follows based on https://www.unaids.org/en/regionscountries/regions definitions and as previously described (5): Sub-Saharan Africa, SSA (Ethiopia, Ghana, Sudan, Nigeria, Congo, Ivory Coast), Eastern Europe and Central Asia, EEU/CA, (Russian Federation, Ukraine, Azerbaijan, Uzbekistan), Western and Central Europe and North America, WCEU/NA (United Kingdom, Romania, Holland, and United States), Latin America, LA (Argentina, Brazil, Peru, Uruguay, Venezuela), Asia and the Pacific, AP (Philippine, Thailand, India), the Middle East and North Africa, ME/NA (Jordan, Tunisia, Yemen, Morocco, Algeria) and Israel. 

  1. Table 2
  2.   It is exceedingly difficult to read as formatted. Since Column 1 is unlabeled and line items below are centered, it is difficult to identify which lines are divisions and which ones are sub-divisions of these. The Reviewer suggests changing the formatting for this Table. As an example, place the “divisions” (i.e., age, sex, place of birth) towards the left margin and right-indent the subdivisions (i.e., female, male) but also aligned towards the left. The same should be done for the other columns. This would direct the line of vision making it easier to understand specially when each division has different N values.

Our reply: Thanks, we improved the visual of this Table as suggested.

  1. Also, it is very confusing the label for columns 2 and 3 indicating “N=808” and “N=120”, respectively. It takes too much time to decipher those numbers have nothing to do with the lines below since all have different “N” numbers. Shouldn’t these “Ns” be written as “n”? aren’t they a sample of the totality of sequences evaluated? Please review and make the corresponding changes to the Table.

Our reply: Our reply: Thanks, was done.

  1. Please check that the location of the “p values” are in the correct line. For example,  in the line “Sex…”, the p value is placed on same line of “female” yet in the text (line 183) it is described as “more males were identified with recombinant viruses (p=0.038).” Observing Table 2 it seems to the Reviewer that the p value refers to “females” and not “males”. Same issue for “ Place of birth…” where the p value is placed in that same line yet in the text (line 186) refers to “Israeli born individuals”  yet in the table “Israel” is placed 4 lines down from where the p value is. Please review for clarity and make corresponding changes to  the Table.

Our reply: Our reply: Thanks, was done.

  1. Figure 3. Consider simplifying the caption as the method and results are found elsewhere in the text.

Our reply: Our reply: Thanks, was done.

" Figure 3. HIV-1 subtypes versus recombinants forms trends by years of diagnosis 2010-2012, 2013-2015, 2016-2018. The proportion, absolute number and trend of HIV-1 subtypes are presented by blue columns and blue dotted line, while the proportion, absolute number and trend of HIV-1 recombinants are presented by red columns and red dotted line."

Reviewer 2 Report

In this work the authors provide an interesting overview of HIV-1 diversity evolution in Israël from 2010 to 2018. They analysed the subtypes and CRFs epidemiology, as well as primary resistance mutation and clinical characteristics across HIV-1 lineages. Such data are useful to monitor and deserve to be analysed and provide to the scientific and public health workers. However, the manuscript presents several critical pitfalls that need to be tackled before publication.

1-      “Pure” subtypes and recombinant forms. The authors often refer to “pure subtypes”, recombinants and secondary recombinants. I would strongly discourage that. Indeed, there is no such things as pure subtypes. They are also recombinants from previous lineages but are simply the earliest one identified in human (but if we had access to previous HIV lineages this would be a different story) and some of them may even be mislabelled (there is an old fight regarding CRF02_AG that is probably a parental strain of subtype G rather than the other way around). Thus, speaking of recombinants of pure lineage is untrue and can be misleading. This is not only a rhetorical question. This highlights the main purpose of such classification: following lineage changes overtime and its association to behavioural changes. Thus, I would strongly recommend the authors to focus and all lineage changes instead of recombinant vs non-recombinant. As an example, I read that A subtype was quite frequent among foreigner’s women diagnosed in Israel and coming from the former soviet union during the same time period in Israel (see PMID 32894102). This may introduce the A6 sub-subtype which is in link with cabotegravir failure. This point (in link with a “pure” lineage) is of huge interest, as are the changes among recombinant forms. A at last, properly described the recombinants, a near full genome sequence should be required but is not reachable at the moment, as stated by the authors. Thus, describing molecular epidemiology instead of all recombinants versus the other lineage would also be more accurate regarding this point.

 2-      The context should be emphasized to help readers that are not aware of the Israel epidemiology. What is the context (in a few words) of the Israel HIV epidemic, the previous data on molecular epidemiology, etc…

 3-      The differences of the patient’s characteristics over years could also be depicted to see those specific from a viral lineage or a time period.

 4-      How were chosen the 410 references from LANL? And how was analysed the phylogenetic tree? Was it only for subtype identification or to decide about RIP and RDP borderline results?

 5-      For the recombinants that are not referenced with a number, those are not described everywhere? Is there any work ongoing for retrieving those patients and describe the full genome sequence?

 6-      The authors should be careful with the conclusions: their results do not prove that the recombinants are an obstacle to containing the local epidemics. Seeing an augmentation of a lineage is not sufficient to evidence that the spread is linked to this lineage.

 7-      Was this study approved by an ethical review board?

Author Response

Response to Reviewer 2 Comments

1.“Pure” subtypes and recombinant forms. The authors often refer to “pure subtypes”, recombinants and secondary recombinants. I would strongly discourage that. Indeed, there is no such things as pure subtypes. They are also recombinants from previous lineages but are simply the earliest one identified in human (but if we had access to previous HIV lineages this would be a different story) and some of them may even be mislabelled (there is an old fight regarding CRF02_AG that is probably a parental strain of subtype G rather than the other way around). Thus, speaking of recombinants of pure lineage is untrue and can be misleading. This is not only a rhetorical question. This highlights the main purpose of such classification: following lineage changes overtime and its association to behavioural changes. Thus, I would strongly recommend the authors to focus and all lineage changes instead of recombinant vs non-recombinant. As an example, I read that A subtype was quite frequent among foreigner’s women diagnosed in Israel and coming from the former Soviet Union during the same time period in Israel (see PMID 32894102). This may introduce the A6 sub-subtype which is in link with cabotegravir failure. This point (in link with a “pure” lineage) is of huge interest, as are the changes among recombinant forms. A at last, properly described the recombinants, a near full genome sequence should be required but is not reachable at the moment, as stated by the authors. Thus, describing molecular epidemiology instead of all recombinants versus the other lineage would also be more accurate regarding this point.

 Our reply: Thank you. We rephrased the word "pure" to "subtype", "non-CRF or “classic subtype”. Regarding non recombinant viruses in Israel (analysis which includes the A6 subtype) we have previously published this data (Wagner Tet al. Viruses. 2021 Dec 31;14(1).) In this report we studied lineages of recombinants and described the overall molecular epidemiology of HIV in Israel.

  1. The context should be emphasized to help readers that are not aware of the Israel epidemiology. What is the context (in a few words) of the Israel HIV epidemic, the previous data on molecular epidemiology, etc…

Our reply: Thanks, we revised the introduction to add background context to out study.

"HIV-1 in Israel started with a subtype-B epidemic among MSM and following by introductions of subtypes C from Sub Saharan Africa (SSA) (mainly by heterosexual transmission) in the 1980s and A from Eastern Europe (mainly by injecting drug users, IDU's) in the 1990s. With the continuous waves of immigration from SSA, the most common subtype in those years was subtype C. (4).

Analysis of the of HIV infection in 2010-2018 in Israel revealed that the epidemic is changing. Overall, 43.9% of the infections were with subtype B, 25.2% with subtype A (22.8% A6 2.4% with A1) and 17.1% with subtype C (5). However, all other infections with none-classic A, B, C subtypes (13.8 %) were not characterized thoroughly".

 3-      The differences of the patient’s characteristics over years could also be depicted to see those specific from a viral lineage or a time period.

Our reply: When time of diagnosis was assessed in patients with recombinant viruses, overall, their characteristics were not different over these years.  We added this clarification to the text of the results (page 4) as follows: "No significant changes in the characteristics of the patients with recombinant viruses (e.g. in the proportion of risk groups, median viral loads and CD4 counts) was observed in the study period." 

 4-      How were chosen the 410 references from LANL? And how was analysed the phylogenetic tree? Was it only for subtype identification or to decide about RIP and RDP borderline results?

Our reply: 410 representative reference sequences that cover all known CRFs and non CRFs of group M were obtained from the Compendium of the Los Alamos database: (www.hiv.lanl.gov/content/sequence/HIV/CRFs/CRFs.html). These were chosen to cover all known group M sequences. Phylogenetic tree was used not only for subtype identification but also to assist in assessing borderline RIP and RDP results.

We revised the text as follows:

"The resultant assembled 253 HIV-1 sequences were aligned (using Ali View version 1.27) (10) with additional 410 representative reference sequences covering all known CRFs and non CRFs group M, obtained from the Compendium of the Los Alamos database: (www.hiv.lanl.gov/content/sequence/HIV/CRFs/CRFs.html)".

And: "Phylogenetic tree was used not only for subtype identification but also to assist in assessing borderline RIP and RDP results"

 5-      For the recombinants that are not referenced with a number, those are not described everywhere? Is there any work ongoing for retrieving those patients and describe the full genome sequence?

Our reply: We are submitting representative batch to Genebank. In the future we will also plan to get full genome sequence for some of these recombinants, especially for those which could not be assigned clearly to known CRF. We added this point to the discussion as follows:

"However, in the future we also plan to obtain full genome sequence for some of these recombinants, especially for those which could not be assigned clearly to known CRF"

  1. The authors should be careful with the conclusions: their results do not prove that the recombinants are an obstacle to containing the local epidemics. Seeing an augmentation of a lineage is not sufficient to evidence that the spread is linked to this lineage.

Our reply: Thank you. Following your remark and our revision we have decided to remove this statement. We have revised as follows:

Abstract: "We concluded that recombinants were mainly detected in Israeli-born MSM and that an increase in the overall proportion of recombinants could be observed in more recent years".

Discussion: "In conclusion, our study documents an increase in the proportion of recombinant viruses among HIV-1 infected individuals particularly in recent years (2016-2018) in Israel. Further, the local circulation of HIV-1 recombinants is mainly by the MSM. These findings highlight the need of strengthening monitoring efforts at the regional and global levels and implementing effective means to reduce the spread of recombinant forms."

 7-      Was this study approved by an ethical review board?

Our reply: Thank you for this remark.  We have added this information in page 4 under "Statistical analysis and ethical approval

                              " The study was approved by the institutional review board of Sheba Medical Center. All data was anonymized, cross-matched and coded as dual-encoding on demand. A consent waiver was obtained for the study. "

Reviewer 3 Report

The article by Wagner and colleagues presents the prevalence of HIV recombinants in Israel during the period from 2010-2018. The authors observe an increase in the prevalence of recombinants and mutations associated with drug resistance. These data are important to report, as they will inform the scientific and medical community of the HIV epidemiology, especially in relation to treatment resistance.

The manuscript is well written, the data is presented clearly and support the conclusions. I only have a few minor comments as outlined below.

Abstract, line 32-35. The authors write that the identification of recombinants may be an obstacle to contain the local epidemic. I think this sentence would improve by including an explanation as to what the obstacle is.

The terms “pure HIV-1” or “pure subtypes”– to indicate non-CRFs – should be rephrased to “non-CRF” or “subtype”, “classic subtype”, “original subtype” or simply without the “pure”.

The abbreviation PLHIV first mentioned (as far as I could see) in line 127 is not introduced. I kindly ask the authors to replace some of the other terms with PLHIV to reduce stigma, for example “infected individuals” on line 62 and “patients” on line 68.

Figure 2. The text in the darker colored background in the pie chart is difficult to read; AB/B, CRF02 etc. Perhaps the text in these parts of the figure could be in white?

Line 163, needs rephrasing.

Lines 181-182. Please indicate which test is used for which dataset.

Line 213. “…. were found to carry recombinant viruses”. Could be rephrased to “ … were found to belong to recombinant viruses”.

Author Response

Response to Reviewer 3 Comments

1.Abstract, line 32-35. The authors write that the identification of recombinants may be an obstacle to contain the local epidemic. I think this sentence would improve by including an explanation as to what the obstacle is.

Our reply: Thank you. Following your remark and our revision we have decided torevise this sentence as follows" We concluded that recombinants were mainly detected in Israeli-born MSM and that an increase in the overall proportion of recombinants could be observed in more recent years"

2.The terms “pure HIV-1” or “pure subtypes”– to indicate non-CRFs – should be rephrased to “non-CRF” or “subtype”, “classic subtype”, “original subtype” or simply without the “pure”.

Our reply: Thanks, we rephrased the word "pure" to "subtype", "on-CRF", or “classic subtype” as suggested throughout the paper.

3.The abbreviation PLHIV first mentioned (as far as I could see) in line 127 is not introduced. I kindly ask the authors to replace some of the other terms with PLHIV to reduce stigma, for example “infected individuals” on line 62 and “patients” on line 68.

Our reply: Thank you. These terms were replaced all through the manuscript.

4.Figure 2. The text in the darker colored background in the pie chart is difficult to read; AB/B, CRF02 etc. Perhaps the text in these parts of the figure could be in white?

Our reply: Thank you. This figure was revised to enable smooth readings.

5.Line 163, needs rephrasing.

Thanks, done.

"Resistance mutation analysis revealed that mutations in protease were very rare accounting for less than 1.5% of the cases".

6.Lines 181-182. Please indicate which test is used for which dataset.

Our reply: Was done, thanks. "Significance of differences was measured using the Chi-squared test (or Fisher's Exact test) for categorical variables like sex, place of birth, transmission groups, TDRM Bennet, HIVdrm ECDC or Mann-Whitney test for continuous variables (age at diagnoses, CD4, HIV-1 – RNA)".

Line 213. “…. were found to carry recombinant viruses”. Could be rephrased to “… were found to belong to recombinant viruses”.

Our reply: Was done, thanks.

Reviewer 4 Report

Wagner and colleagues report an analysis of HIV-1 recombinants in Israel from patients diagnosed in 2010-2018, based in partial pol sequences. They analyze HIV-1 genetic forms to correlate with transmission risk groups, geographic origin of patients and antiretroviral resistance transmission. This report complements a recently published one, by the same group, about HIV-1 molecular epidemiology and transmitted drug resistance in the same cohort. However, in the previously published article, the authors only focused in analyzing pure subtypes, while recombinants are analyzed in the present manuscript.

 Main comments:

 In the analysis of recombinants is important to analyze separately Circulating Recombinants Forms (CRFs) and Unique Recombinant Forms (URFs), because circulation of some CRFs (such as CRF01_AE and CRF02_AG) is comparable to circulation of pure subtypes. Also in the referenced paper of Hemelaar (ref.1) CRFs and URFs are analyzed separately.

 It is also epidemiologically relevant to identify clusters of recombinants, which could be putative new CRFs, with some advantages in transmission. In order to reinforce this analysis, Figure S1 (the phylogenetic tree) should appear as one of the figures in the article body (not supplementary).

 About the tree, transmission clusters should be identified and accompanied by a branch support value in the cluster node. In addition, in the methods section, it is missed the program used to construct the tree. The criteria to select sequences from public databases to be included in the phylogenetic tree are not explained.

 Recombination patterns should be shown in a figure, or at least, mosaics of any recombinant cluster identified.

 This report is a description of the recombinants identified; however, it lacks a deeper analysis to differentiate which recombinants are circulating in Israel, and which ones are linked to the countries of origin of individuals.

 Sequences should be deposited in GenBank, at least, a proportion of each genetic form.

 It is not clear the reason to choose a 50% random sample for comparisons, while the previous study includes all the patients. My recommendation is to include all the patients.

Minor comments and typographical errors:

Title: HIV-1 (not HIV).

Page 1, line 41: HIV-1 inter (space).

Page 2, line 72: diagnosed in 2010-… (lowercase).

Page 2, line 75: these sequences “are intersubtype recombinants” (instead of “contain CRFs”).

Page 3, line 129: B/D recombinants should be well characterize in phylogenetic trees, as B and D subtypes are very similar to de discriminated by several programs.

Page 5, line 159: CRF02_AG (instead of CRF_02AG).

Page 7, line 217: CRF01_AE (instead of CRF02_AE).

Page 9, lines 320-321: Tel Aviv, Israel (uppercase).

Author Response

Response to Reviewer 4 Comments

 Main comments:

  1. In the analysis of recombinants is important to analyze separately Circulating Recombinants Forms (CRFs) and Unique Recombinant Forms (URFs), because circulation of some CRFs (such as CRF01_AE and CRF02_AG) is comparable to circulation of pure subtypes. Also, in the referenced paper of Hemelaar (ref.1) CRFs and URFs are analyzed separately.

Our response: Thank you. We were reluctant to separate CRF from URF based on the partial sequences only. However, most of the viruses were indeed CRF and only <10% (based on our initial assessment and without having the whole HIV-1 sequenced) were Unique Recombinant Forms (URFs).

Therefore, due to the low proportion of suspected URFs, and the obstacle to predict exactly the true number of URFs, we decided not to differentiate between CRFs and URFs in this report and to refer to all these sequences as recombinants as opposed to subtypes for the other non-recombinant viruses.

This change was implemented throughout the paper

  1. It is also epidemiologically relevant to identify clusters of recombinants, which could be putative new CRFs, with some advantages in transmission. In order to reinforce this analysis, Figure S1 (the phylogenetic tree) should appear as one of the figures in the article body (not supplementary).

Our response: Thank you, following your remark we identified clusters of recombinants and marked them with Roman letters in the figure (Fig 2). We moved the figure to the main text and updated the caption of this figure accordingly as follows:

 "Main clusters were determined according to the posterior probabilities >0.8 and number of individuals (>2) in each cluster. Main clusters were named with the Roman letters and indicated with the blue splines. Posterior probabilities of each cluster were: I. 0.97; II. 0.97; III. 0.99; IV. 0.9; V. 0.88; VI. 0.97; VII. 1; VIII. 1; IX. 0.85."

  1. About the tree, transmission clusters should be identified and accompanied by a branch support value in the cluster node.

Our response: Thank you, we identified potentially transmitted clusters of recombinants and accompanied them by a branch support value (see previous reply).

  1. In addition, in the methods section, it is missed the program used to construct the tree.

Our response: Thank you, we added the name of the program "All these sequences were also explored with the maximum-likelihood phylogenetic method with Mega X software ( http://www.megasoftware.net),using the General time-reversible model, with the proportion of invariable sites and gamma plus invariant sites-distributed rate heterogeneity (GTR+G+I model) based on the results of JModelTest (11) version 2.1.9. Maximum likelihood phylogenetic tree was constructed with branch support assessed from 1000 bootstrap replicates and visualized with Fig Tree version 1.4.4 (http://tree.bio.ed.ac.uk/software/figtree/).”

      Line 88 (Methods).

  1. The criteria to select sequences from public databases to be included in the phylogenetic tree are not explained.

Our response: According to the Rhee SY and Shafer RW recommendations (page 8)

(Rhee S-Y, Shafer RW. Geographically stratified HIV-1 group M pol subtype and circulating recombinant form sequences. Sci Data. 2018 Jul 31; 5:180148.) we included a set of reference sequences containing a CRF and also the non-CRF sequences. See revised paragraph below:

"First, partial protease (PR, codons 4-99) and reverse transcriptase (RT, codons 38-247) sequences from samples suspicious to contain HIV-1 recombinant forms (single sequence per patient, n=270) were assembled using an R script. Sequences shorter the 918 nucleotides (n=17) were excluded. The resultant assembled 253 HIV-1 sequences were aligned (using Ali View version 1.27) (10) with additional 410 representative reference sequences covering all known CRFs and non CRFs group M, obtained from the Compendium of the Los Alamos database (www.hiv.lanl.gov/content/sequence/HIV/CRFs/CRFs.html). Recombinants were defined with RIP (Recombinant Identification Program (available at www.hiv.lanl.gov/content/sequence/RIP/RIP.html) and with RDP (Recombination Detection Program) version 5.5 (11) . CRFs were determined according to guidelines available at (www.hiv.lanl.gov/content/sequence/HIV/CRFs/crfs.comp). All these sequences were also explored with Mega X software ( http://www.megasoftware.net) using the maximum-likelihood phylogenetic method and the General time-reversible model, with the proportion of invariable sites and gamma plus invariant sites-distributed rate heterogeneity (GTR+G+I model) based on the results of JModelTest (12) version 2.1.9. Maximum likelihood phylogenetic tree was constructed with branch support assessed from 1000 bootstrap replicates and visualized with Fig Tree version 1.4.4 (http://tree.bio.ed.ac.uk/software/figtree/). Phylogenetic tree was used not only for subtype identification but also to assist in assessing borderline RIP and RDP results. Following analysis of all sequences with these three tools, sequences with no evidence of recombination (n=40) were removed from further analysis, yielding a final set of 213 HIV-1 recombinants. The results of these analyses and the detailed list of all 213 sequences of HIV-1 recombinants is provided in Table S1.. Recombination breakpoints (according to HXB2 genome positions) were obtained using RDP5 and Simplot v3.5.1 software. Mosaic patterns were generated using the Recombinant HIV-1 Drawing Tool (https://www.hiv.lanl.gov/content/sequence/DRAW_CRF/recom_mapper.html.  "

  1. Recombination patterns should be shown in a figure, or at least, mosaics of any recombinant cluster identified.

Our reply: Thank you, we added the mosaic pattern of main recombinant clusters as Figure S2 in the supplementary section.

"Figure S2. Mosaic pattern of main HIV-1 recombinant clusters of the pol region. Protease (PR) (2262-2547 bp) and partial reverse transcriptase (RT) gene (2662–3290 bp) are shown. Recombination breakpoints (according to HXB2 genome positions) were obtained using RDP5 and Simplot v3.5.1 software. Each color stands for different HIV-1 subtypes. A representative sequence of each of the main clusters in the phylogenetic tree which are labeled with roman letters presented here and are labeled with the same roman letters used in the Figure 2. The mosaic map was generated using the Recombinant HIV-1 Drawing Tool (https://www.hiv.lanl.gov/content/sequence/DRAW_CRF/recom_mapper.html)".

  1. This report is a description of the recombinants identified; however, it lacks a deeper analysis to differentiate which recombinants are circulating in Israel, and which ones are linked to the countries of origin of individuals.

Our reply: Phylogenetic analysis demonstrated that most of PLHIV infected outside Israel were not clustered. Also, there were no significance difference in characteristics of PLHIV who infected in Israel and were part of any cluster and between those who were not in any cluster. This finding does not support the possibility of some pattern of the transmission of the infection, however, does not cancel it.

To address this issue we have revised the results section as follows

"The majority of PLHIV harboring recombinant viruses (61%) were born in Israel. A travel history to HIV-1 endemic countries was reported for few case (7/127 Israelis). Indeed, for these individuals as well as for those born in such endemic areas, association between the HIV-1 recombinant and the country of infection could be identified. Thus, PLHIV born in or travelled to Africa were most often infected with CRF02_AG (16/25, 64%, p<0.001). CRF01_AE subtype characterized HIV-1 positives who were infected in Asia and the Pacific (9/12, 75%, p<0.001). Most of those with B/F1virus were infected in Latin America (4/6, 66.7%, p<0.001).  Phylogenetic analysis demonstrated that most (70%) of PLHIV infected outside Israel were not clustered. Also, there were no significant differences in the characteristics of PLHIV that were clustered and PLHIV who were not in any cluster".

In the discussion we included the following:

"Most of those newly infected with the recombinant virus (66.1%) were PLHIV born in Israel. For a minority of these individuals, information on infection in endemic regions was available and could partially explain the introduction of such HIV-1 recombinants in Israel. Indeed, data on country of infection is important to understand the epidemiology and spread of HIV recombinants for regional and global surveillance. Active circulation of HIV-1 recombinants in Israel could not be determined as the characteristics of clustered individuals were not significantly different to those who were not in any cluster".

  1. Sequences should be deposited in GenBank, at least, a proportion of each genetic form.

Our reply: Thank you, we will deposit representative sequences soon.

  1. It is not clear the reason to choose a 50% random sample for comparisons, while the previous study includes all the patients. My recommendation is to include all the patients.

Our reply: We provided a 50% random sample for comparisons in order to overcome a possible selection bias in cases which are sequenced, as we learnt that sequencing was done to only ~50% of cases, and the reason for this is unknown. Using random selection we thought we will have a better representation of the data in Israel. The total number of cases compared here was >900.

Minor comments and typographical errors:

Title: HIV-1 (not HIV).

Our reply: Thanks, corrected

Page 1, line 41: HIV-1 inter (space).

Our reply: Thanks, corrected

Page 2, line 72: diagnosed in 2010-… (lowercase).

Our reply: Thanks, corrected

Page 2, line 75: these sequences “are intersubtype recombinants” (instead of “contain CRFs”).

Our reply: Thank you very much, done.

Page 3, line 129: B/D recombinants should be well characterize in phylogenetic trees, as B and D subtypes are very similar to de discriminated by several programs.

Our reply: Thank you very much, done.

Page 5, line 159: CRF02_AG (instead of CRF_02AG).

Our reply: Thank you very much, done.

Page 7, line 217: CRF01_AE (instead of CRF02_AE).

Our reply: Thanks, done.

Page 9, lines 320-321: Tel Aviv, Israel (uppercase).

Our reply: Thank you very much, done.

Round 2

Reviewer 1 Report

Manuscript ID:   viruses-1835931

Wagner et al.

HIV Recombinants in Israel, 2010-2018                                   

In this manuscript, Wagner et al. report the characterization of the diversity and epidemiology of HIV-1 inter-subtype recombinants in Israel from 2010 to 2018 among newly diagnosed PLHIV. Multiple types of CRFs and various complex CFRs were identified amongst the studied individuals. The study provides information regarding the evolution of HIV-1 in Israel, noting the increased frequency of recombinant forms (CRFs) over the studied period in the newly diagnosed population.

The Reviewer thanks the Authors for thoroughly responding to our comments and critiques. The Reviewer finds that these have been addressed satisfactorily and therefore finds the present manuscript ready to be considered for publication.

Author Response

Thank you 

Reviewer 4 Report

Response to authors’ responses to previous comments:

11.      It is clearly stablished that the study is based on partial pol sequences. Therefore, the classification is common to define either subtypes or recombinants. Some subtypes could be recombinants after sequencing other genome fragments, in the same way that some CRFs could be URFs with further sequencing. The same occurs in the referenced paper from Hemelaar et al., where a large proportion of sequences used are partial pol genomes, and there, it is relevant to distinguish between CRFs and URFs. The main point here is that some CRFs with wide circulation should not be epidemiologically differentiated to some subtypes. In my opinion, a plain comparison between recombinants and non-recombinants, without distinction of CRFs and URFs, is not a good design to obtain practical conclusions. Moreover, figure 4 would be much more informative if the main genetic forms (A, B, C, CRF01_AE, CRF02_AG, other) were indicated in each bar (subtypes/recombinants) in each period.

I do not appreciate that the proposed change was implemented in the text, as the authors responded here.

22.      The inclusion of a phylogenetic tree as a main figure is a good improvement to this manuscript. However, there are important flaws in this figure to be published:

a.       The selection of references: References included are not adequate to get conclusions from the phylogenetic tree. A tree must be well structured by including references of subtypes, CRFs related with the samples analysed, and related sequences from BLAST searches. Inclusion of 410 references without a clear selection related to the samples analysed, and without indication of the location of references (pure subtypes and CRFs), makes this tree very difficult to structure.

b.       Description of colours, and support values of each cluster should be indicated in the figure, because in the figure legend results confusing.

c.       As far as I know, to get posterior probabilities it is necessary a Bayesian analysis, which is not implemented in MegaX.

33.       Identification of clusters should be accompanied by a description of their characteristics in the text or in a table, with the aim of searching whether any cluster is expanding in a transmission group, as well as to identify putative new CRFs.

44.       Done, the program to construct the tree is specified.

55.      The criteria to select references is not adequate for the purpose of this study, as I mentioned before (point 2.a).

66.      Recombination patterns of clusters classified as CRFs should not be included in figure S2, as those recombination patterns are already described in the CRF (i.e. for CRF02_AG in clusters I, II and III). What I meant was to show new recombination patterns in clusters, which could be putative new CRFs. Also, I do not find robust enough the characterization of cluster V (CRF02_AG/A4), which could be epidemiologically relevant, as it is quite numerous (14 infections) and a resistance mutation could be transmitted with it.  So far, subsubtype A4 has not been described outside DR Congo and the fragment of subsubtype A4 is too short. Mosaic patterns for short fragments need to be supported by phylogenetic trees. Also, sequences from this cluster have not been deposited in GenBank.

77.     This point has been improved, but not enough, as the authors have done a general analysis comparing clustering and no clustering, but I was asking for a deeper analysis to identify clusters and genetic forms with higher circulation in the population to address the causes of the increasing of recombinants along time.

88.      Sequences deposited in GenBank appear in table S2. However, I do not find it mentioned in the main text. Moreover, sequences from the clusters identified (at least two of each) should be deposited in public databases in order to help identification of their expansion. Also, in table S2 a column with the cluster ID should be included.

99.  I still think that with this 50% selection, some possibly relevant information is missed.

110.   Page 9, lines 320-321: Tel Aviv, Israel (uppercase): not done.

Despite my rejection, I would like to encourage the authors to continue this line of HIV-1 molecular epidemiology research, with a better design and refining the presentation of results. This type of studies, for sure, contribute to a better understanding of global HIV-1 epidemiology and are helpful to monitor local epidemics. However, the manuscript still has serious flaws to be published, and a deeper analysis of the recombinants in expansion and its implications is required.

Author Response

Comments and Suggestions for Authors

Response to authors’ responses to previous comments:

  1. It is clearly stablished that the study is based on partial pol sequences. Therefore, the classification is common to define either subtypes or recombinants. Some subtypes could be recombinants after sequencing other genome fragments, in the same way that some CRFs could be URFs with further sequencing. The same occurs in the referenced paper from Hemelaar et al., where a large proportion of sequences used are partial pol genomes, and there, it is relevant to distinguish between CRFs and URFs. The main point here is that some CRFs with wide circulation should not be epidemiologically differentiated to some subtypes. In my opinion, a plain comparison between recombinants and non-recombinants, without distinction of CRFs and URFs, is not a good design to obtain practical conclusions. Moreover, figure 4 would be much more informative if the main genetic forms (A, B, C, CRF01_AE, CRF02_AG, other) were indicated in each bar (subtypes/recombinants) in each period.

I do not appreciate that the proposed change was implemented in the text, as the authors responded here.

Our reply: We understand the CRFs with wide circulation are different from other sequences which are more rare. We have revised the paper so that it now addresses the prevalence and main characteristics of PLHIV infected with either CRFs or URFs. Figure 4 was also revised as suggested

  1. The inclusion of a phylogenetic tree as a main figure is a good improvement to this manuscript. However, there are important flaws in this figure to be published:
  2. The selection of references: References included are not adequate to get conclusions from the phylogenetic tree. A tree must be well structured by including references of subtypes, CRFs related with the samples analysed, and related sequences from BLAST searches. Inclusion of 410 references without a clear selection related to the samples analysed, and without indication of the location of references (pure subtypes and CRFs), makes this tree very difficult to structure.
  3. Description of colours, and support values of each cluster should be indicated in the figure, because in the figure legend results confusing.
  4. As far as I know, to get posterior probabilities it is necessary a Bayesian analysis, which is not implemented in MegaX.

Reply: For our phylogenetic analysis we took all available recombinant sequences and representatives of all pure subtypes from LANL. These include sequences obtained by BLAST analysis that were related to the samples analyzed herein. Regarding the description of colours in the figure - we checked again and ensured that all needed explanations are included in the caption of the figure. Posterior possibilities were written by mistake, we changed to bootstrap values.

  1. Identification of clusters should be accompanied by a description of their characteristics in the text or in a table, with the aim of searching whether any cluster is expanding in a transmission group, as well as to identify putative new CRFs.

Reply: New table (s3) with description of the main clusters is provided in supplement material

  1. Done, the program to construct the tree is specified.

Reply: Thanks

  1. The criteria to select references is not adequate for the purpose of this study, as I mentioned before (point 2.a).

Reply: See our previous reply

  1. Recombination patterns of clusters classified as CRFs should not be included in figure S2, as those recombination patterns are already described in the CRF (i.e. for CRF02_AG in clusters I, II and III). What I meant was to show new recombination patterns in clusters, which could be putative new CRFs. Also, I do not find robust enough the characterization of cluster V (CRF02_AG/A4), which could be epidemiologically relevant, as it is quite numerous (14 infections) and a resistance mutation could be transmitted with it. So far, subsubtype A4 has not been described outside DR Congo and the fragment of subsubtype A4 is too short. Mosaic patterns for short fragments need to be supported by phylogenetic trees. Also, sequences from this cluster have not been deposited in GenBank.

Reply: OK, we updated the Figure as requested (Fig S2). We also prepared a tables that summarizes all clusters and includes the resistance mutations related to cluster V. Regarding this cluster, indeed the fragment related to A4 is short, however, these sequences could not be designated as CRF02_AG. We are now in the process of depositing all sequences to GeneBank.

  1. This point has been improved, but not enough, as the authors have done a general analysis comparing clustering and no clustering, but I was asking for a deeper analysis to identify clusters and genetic forms with higher circulation in the population to address the causes of the increasing of recombinants along time.

Reply: We have addressed this issue in the revised manuscript (Revised Table S1 and new Table S3, cluster analysis)

  1. Sequences deposited in GenBank appear in table S2. However, I do not find it mentioned in the main text. Moreover, sequences from the clusters identified (at least two of each) should be deposited in public databases in order to help identification of their expansion. Also, in table S2 a column with the cluster ID should be included.

Reply: We revised the main text and mention GeneBank, We also referred to the GeneBank number in caption of Table S1.

  1. I still think that with this 50% selection, some possibly relevant information is missed.

Reply: OK, but we believe that the overall number of cases included in this comparison is still rather high to enable drawing conclusions

  1. Page 9, lines 320-321: Tel Aviv, Israel (uppercase): not done.

Reply: Thanks, corrected

Despite my rejection, I would like to encourage the authors to continue this line of HIV-1 molecular epidemiology research, with a better design and refining the presentation of results. This type of studies, for sure, contribute to a better understanding of global HIV-1 epidemiology and are helpful to monitor local epidemics. However, the manuscript still has serious flaws to be published, and a deeper analysis of the recombinants in expansion and its implications is required.